# Novel Necroptosis-Related Gene Signature for Predicting Early Diagnosis and Prognosis and Immunotherapy of Gastric Cancer

**DOI:** 10.3390/cancers14163891

**Published:** 2022-08-11

**Authors:** Xiaozhu Zhou, Baizhuo Zhang, Guoliang Zheng, Zhen Zhang, Jiaoqi Wu, Ke Du, Jing Zhang

**Affiliations:** 1Department of Pharmacology, College of Pharmacy, China Medical University, Shenyang 110042, China; 2Department of Gastric Surgery, Cancer Hospital of China Medical University (Liaoning Cancer Hospital and Institute), Shenyang 110042, China

**Keywords:** necroptosis, gastric cancer, bioinformatics, diagnosis, prognosis, immunotherapy, FAP, machine learning

## Abstract

**Simple Summary:**

Necroptosis plays an important role in the occurrence and development of many cancers. MLKL is an important component of necroptosis, and has been proved to be closely related to the prognosis of gastric cancer (GC). We determined an early diagnosis (FAP, CCT6A) and prognosis risk score (ZFP36, TP53I3, FAP, CCT6A) model of necroptosis-related genes (NRGs) in GC. Two models, respectively, can effectively predict the occurrence of GC and the prognosis of GC patients. The association between the prognostic risk score and the response to immunotherapy and immune checkpoint inhibitors (ICIs) was also analyzed. FAP was also identified as the core gene in the two models, and the relationship between its expression in GC and ICIs was analyzed. This discovery is the first time that NRGs were combined with immunotherapy for GC and provides a new target for immunotherapy for GC and a more accurate treatment scheme for GC patients.

**Abstract:**

Necroptosis is a kind of programmed necrosis, which is different from apoptosis and pyroptosis. Its molecular mechanism has been described in inflammatory diseases. Gastric cancer (GC) is one of the most common malignancies worldwide with the third highest mortality. However, the role of necroptosis in the occurrence and progression of GC remains largely unexplored. Therefore, we investigated necroptosis-related genes (NRGs) by analyzing public transcriptomic data from GC samples. Our results indicate that 83 of 740 NRGs are dysregulated in GC tissues. Next, we identified necroptosis-associated early diagnosis and prognostic gene signatures for GC using machine learning. 2-NRGs (CCT6A and FAP) and 4-NRGs (ZFP36, TP53I3, FAP, and CCT6A), respectively, can effectively assess the risk of early GC (AUC = 0.943) and the prognosis of GC patients (AUC = 0.866). Through in-depth analysis, we were pleasantly surprised to find that there was a significant correlation between the 4-NRGs and GC immunotherapy effect and immune checkpoint inhibitors (ICIs), which could be used for the evaluation of immunosuppressants. Finally, we identified the core gene FAP, and established the relationship between FAP and ICIs in GC. These findings could provide a new target for immunotherapy for GC and a more effective treatment scheme for GC patients.

## 1. Introduction

Gastric cancer (GC) is one of the most common malignant tumors in the world; its global incidence ranks fifth among malignant tumors and its mortality ranks third. GC is indeed a malignant tumor that seriously threatens human health and even life. It is also one of the most common malignant tumors in our country. There are 478,508 new cases in China, accounting for 10.5% of all malignant tumors and ranking third; additionally, there were 714,669 deaths in the same period, accounting for 23.8% of all malignant tumor mortality, and, again, ranking third [1]. GC can be divided into early GC and advanced GC (intermediate and advanced GC) according to the depth of cancer tissue invasion [2]. The cause of the disease is also related to eating habits and stomach-related diseases [3]. Early GC symptoms are not obvious, such as pantothenic acid, loss of appetite, and so on. In addition, gastroscopy gives many patients a certain degree of resistance, and it is easy to miss the best treatment time, thus developing into advanced stage, which has a high incidence and poor prognosis of cancer [4]. Therefore, these data suggest that the search [5,6,7] for effective biomarkers is crucial for the early diagnosis and prognosis of GC patients.

Necroptosis [8] is a type of programmed cell death, and it is also the most typical programmed necrosis. It is mediated by various cytokines or pattern recognition receptors (PRRs), independent of caspase; among them, serine-threonine kinase 1 (RIPK1), serine-threonine kinase 3 (RIPK3) [9], and mixed-lineage kinase domain-like protein (MLKL) are key proteins that play important roles in necroptosis [10]. Recent studies have shown that RNA viruses and various bacterial pathogens can activate Z-DNA-binding protein 1 (ZBP1) or toll-like receptor (TLR) to trigger necroptosis [11]. In addition, related literature report that human inflammation-related diseases, such as inflammatory bowel disease and neuroinflammation, are closely related to necroptosis [12]. There have been new reports correlating inflammatory diseases with necroptosis and have explored the role of necroptosis from in vitro studies to pathophysiological studies. Other related studies have shown that necroptosis is also related to tumor anti-immunity, and plays a significant role in the occurrence and development of many immune system-related diseases [13]. Likewise, when cellular necroptosis is out of balance, necrotic cells expel their contents, which stimulates the surrounding inflammatory and immune responses, leading to the development or even worsening of cancer. Therefore, necroptosis plays a crucial role in the initiation and progression of cancer. However, only a few studies [5] have investigated the association of abnormal necroptosis-related gene expression with prognosis [14] and early diagnosis in GC patients. Moreover, these studies [5] only reported the prognostic characteristics of NRGs in gastric adenocarcinoma, but did not report the relevant early diagnosis characteristics, and the performance of the prognostic risk score was also low. Therefore, the study of NRGs in GC also deserves further exploration.

In recent years, the clinical application of immune checkpoint inhibitors (ICIs) has been increasing and has become a major choice for GC patients [15]. Clinical drugs used to treat GC patients with ICIs include anti-PD-L1, anti-CTLA-4, and anti-pd-1 drugs [16]. However, ICIs are not effective for all patients [17,18,19], with certain limitations. Therefore, finding potential therapeutic targets is crucial.

This study systematically assessed the role of necroptosis-related genes (NRGs) in GC patients using a retrospective dataset; a prognostic risk assessment model and a nomogram with risk and clinical characteristics were constructed. The association of signaling pathway enrichment, immune microenvironment (TME), immune cell infiltration, therapeutic response, immune checkpoints, and ICIs with prognostic risk scores, and the relationship between CCT6A, FAP, ZFP36, and TP53I3 and drug sensitivity and resistance to chemotherapeutic drugs were also explored. Based on the prognostic risk score model, an early diagnosis model was constructed and used to predict the early diagnosis ability of GC patients. Finally, FAP was identified as the core gene, and its relationship with the TME, immune cells, immune checkpoints, and ICIs was studied. A comprehensive understanding of the role of NRGs and their targets will help clinicians predict the early onset risk of GC and the prognostic risk of GC patients, and thus provide more precise treatment options for GC patients.

## 2. Materials and methods

### 2.1. Acquisition and Preprocessing of the Datasets

The GC transcriptome dataset and clinical information (449) were obtained from the Cancer Genome Atlas (TCGA) official website (https://portal.gdc.cancer.gov/ accessed on 25 March 2022), and a total of 413 samples (32 normal samples, 381 tumor samples) were obtained (Table 1). The GC gene mutation and copy number variation (CNV) dataset was obtained from the UCSC Xena official website (https://xenabrowser.net/datapages/ accessed on 25 March 2022), and a total of 440 samples were obtained. The GC-related transcriptome dataset (accession number: GSE26899, platform: GPL6947) was downloaded from the Gene Expression Omnibus (GEO) database (https://www.ncbi.nlm.nih.gov/ accessed on 25 March 2022), and a total of 12 normal human samples and 96 tumor samples were obtained (Table 1). Perl software (perl-5.32.1.1) and the R programming language ((version 4.1.3, The R language was originally developed by Ross Ihaka and Robert Gentleman at the University of New Zealand; Auckland, New Zealand) (https://www.r-project.org/ accessed on 25 March 2022)) were used to perform a series of bioinformatic analyses. Through the official website of Bioconductor (https://bioconductor.org/packages/release/bioc/html/ accessed on 25 March 2022), R package “limma” is normalized and converted into a standardized dataset by Log 2.

### 2.2. Acquisition of Human Necroptosis-Related Gene Set

A total of 13 NRGs were collected from previous literature reports [20,21,22,23,24]; 159 NRGs were collected from the Kyoto Encyclopedia of Genes and Genomes (KEGG) (https://www.kegg.jp/ accessed on 25 March 2022) using the keyword “necroptosis”; and 614 NRGs were collected from the Human Gene Database (https://www.genecards.org/ accessed on 25 March 2022) with the keyword “necroptosis gene”. Finally, a total of 781 NRGs were obtained using the three methods.

### 2.3. Gene Annotation and NRGs Differential Analysis

Perl is used for gene annotation for all gene sets that require ID conversion so that gene IDs are converted into gene names to obtain a more accurate gene expression matrix. At the same time, the differentially expressed necroptosis-related genes (DE-NRGs) were screened using the R package “limma”, with FDR < 0.05 and |log2FC| > 1 as the selection criteria, and the R packages “gplots” and “pheatmap” were used to plot DE-NRG’s volcano map and heatmap.

### 2.4. Gene Mutation, CNV and Biological Function Analysis of DE-NRGs

We used Perl software and “maftools” on the Bioconductor official website to perform gene analysis on DE-NRGs mutation correlation and obtain visualization results, and then used Perl software and R package “RCircos” to perform CNV analysis and visualize the results. Furthermore, to investigate the biological functions of DE-NRGs, we performed various functional enrichment analyses, including KEGG and Gene Ontology (GO). We used the R package “org.Hs.eg.db” on the Bioconductor official website to convert the IDs of DE-NRGs, and the R packages “GOplot”, “enrichplot”, and “ggplot2” to perform GO and KEGG enrichment analysis and visualize the results.

### 2.5. Construction and Validation of DE-NRGs Prognostic Risk Scoring Model

The dataset was firstly divided into a training set (Train) group and a test set (Test) group according to 6:4 using the R package “caret”. Univariate Cox proportional hazard regression analysis was performed in the Train group using the R package “survival” to screen out the DE-NRGs that were significantly associated with prognosis in the TCGA GC dataset, and *p* < 0.05 was used as the screening criteria. The risk ratio (HR) and 95% confidence interval (CI) were tested by the Wilcoxon rank test. Furthermore, the R packages “glmnet” and “survival” were used to perform least absolute shrinkage and selection operator (LASSO) regression analysis, and the prognostic genes meeting the criteria were screened out. Multivariate Cox proportional hazard regression was performed using the R package “survival” and “survminer” to further screen the prognostic-related genes. The HR and 95% CI were tested by the Wilcoxon test, and the prognostic risk score model was constructed. The risk score for each patient in the Train group was calculated using the following formula: risk score = eSum (expression of each gene × corresponding coefficient).

The patients in the Train group were divided into low-risk and high-risk groups according to the risk score, with the median value as the cut-off value. The difference in overall survival (OS) between patients in the low-risk and high-risk groups was analyzed using the R package “survival” and the Kaplan-Meier (K-M) survival curve was drawn using the Wilcoxon test. The R packages “timeROC”, “ROCR”, and “survival” were used to plot receiver operating characteristic (ROC) curves for 1, 3, and 5 years, and the R packages “limma” and “scatter plot3d” were used to perform principal component analysis (PCA). The expression of genes in the risk score model was explored in GC and normal tissues using the GEPIA online tool (http://gepia.cancer-pku.cn/ accessed on 10 April 2022). Finally, the genes in the model were verified by immunochemistry from the Human Protein Atlas (http://www.proteinatlas.org/ accessed on 10 April 2022) (HPA) database. In addition, the predictive ability of the model was tested by the Test group and the full set of TCGA.

### 2.6. Independent Prognosis of Risk Score and Correlation Analysis of Clinical Characteristics

Univariate Cox proportional hazards regression and multivariate Cox proportional hazards regression analysis were performed by using the R package “survival”; age, sex, tumor subtype, pathological stage, histological grade, and risk scores were considered as covariates; and *p* < 0.05, HR > 1, and 95%CI were used as screening criteria. A concordance index (C-index) was constructed using the R packages “survival” and “rms” to evaluate the independent prognostic predictive power of risk scores and clinical features. Finally, the chi-square test was used to compare the differences in clinicopathological characteristics between different risk groups, and the R package “pheatmap” was used to draw a heatmap of the correlation between prognostic risk scores and clinical characteristics.

### 2.7. Nomogram Construction and Analysis

Based on the results of multivariate Cox proportional hazards regression analysis, the R packages “survival” and “regplot” were used to construct a nomogram that predicted patient survival at 1 year, 3 years, and 5 years, and then the patient’s risk score information was labeled for clinical use. The R package “RMS” was used to plot the calibration curves for 1 year, 3 years, and 5 years, and the R packages “survival”, “survminer”, “timeROC”, and “ggDCA” were used to plot the multi-index ROC curves and decision curve analysis (DCA) curves.

### 2.8. GSEA Enrichment Analysis of Risk Scores

GSEA software ((GSEA_version 4.2.3, GSEA was originally developed by Broad Institute; Boston, MA, USA)) was used for GSEA analysis with the criteria of *p* < 0.05, FDR < 0.05, and 1000 genome permutations in each analysis.

### 2.9. Correlation Analysis between Risk Score and TME

First, we used the R package “limma” and “estimate” to perform 1000 permutations through the ESTIMATE algorithm to calculate the Stromalscore, Immunescore, and ESTIMATEScore in TME in GC samples. Then, the R packages “limma”, “reshape2”, and “ggpubr” were used to evaluate the relationship between them and the risk score; the R packages “ggplot2”, “ggpubr”, and “ggextra” were used for correlation analysis through the Spearman correlation test to evaluate their correlation with risk score. *p* < 0.05 was the screening standard.

### 2.10. Correlation Analysis of Risk Score with Immune Cell Infiltration and Immune Checkpoints

CIBERSORT and preprocessCore algorithms were used to perform single-sample enrichment analysis (ssGSEA). The R packages “limma”, “GSEABase”, and “GSVA” were used to evaluate the association between the scores of infiltration of 16 immune cells and 13 functions and the risk score, and the boxplot was drawn with *p* < 0.05 as the screening criterion. Then, we used the R packages “limma”, “scales”, “ggplot2”, “reshape2”, and “ggpubr” via XCELL, TIMER, QUANTISEQ, MCPCOUNTER, EPIC, CIBERSORT, and CIBERSORT algorithms, using the Spearman correlation test, to analyze the correlation between immune cells and different risk groups, with *p* < 0.05 as the screening criterion. We collected common immune checkpoint-related genes from literature reports [25,26,27], and then used the R packages “limma”, “ggplot2”, “reshape2”, and “ggpubr” to evaluate the association between immune checkpoint-related genes and risk scores for different risk groups. *p* < 0.05 was the screening standard.

### 2.11. Correlation Analysis of Risk Score with Immunotherapy Response and ICIs

Tumor Immune Dysfunction and Exclusion (TIDE) (http://tide.dfci.harvrd.edu/ accessed on 7 May 2022) is an online tool that can be used to calculate immunotherapy response scores. TIDE online tool was used to calculate the immune dysfunction score, the immune exclusion score, the cancer associated fibroblast (CAF) score, and the TIDE score of the normalized TCGA expression profile; we then used the R packages “limma” and “ggpubr”, respectively, to evaluate them and risk scores, with *p* < 0.05 as the screening standard. Then, the clinical data of the TCGA expression matrix and immune checkpoint blockers (anti-PD1 and anti-CTLA4) were downloaded from the Cancer Immunome Atlas (TCIA) website (https://tcia.at/home accessed on 7 May 2022), using the R packages “limma” and “ggpubr” to assess the association between the use of anti-PD1 and anti-CTLA4 blockers alone or in combination in different risk groups, with *p* < 0.05 as the screening criterion.

### 2.12. Drug Susceptibility and Resistance Analysis of Genes in Risk Scoring Models

The GSCA online tool (http://bioinfo.life.hust.edu.cn/GSCA/#/ accessed on 7 May 2022) was used to explore the relationship between genes in the risk scoring model and drug sensitivity and resistance.

### 2.13. Correlation Analysis of Gene Expression and Pathological Stage in Risk Scoring Model and Construction of Early Diagnosis Model

The UALCAN database (http://ualcan.path.uab.edu accessed on 7 May 2022) was used to analyze the association between the expression of genes in the risk score model, which include normal tissues of GC patients, and the expression in tumor tissues of GC patients who are in different pathological stages. According to their analysis results, the genes whose expression in normal tissues of GC patients is correlated with the expression in tumor tissues, and where the pathological stages of GC patients are stage I and stage II, are determined. Subsequently, we selected the GSE26899 chip, and the samples of clinical pathology stage I with stage II and normal were selected according to their clinical information. The target gene expression matrix was extracted using Sanger Box software ((Sanger Box version 1.0.9, The Sanger Box software was originally developed by Hangzhou Mugu Technology Co., LTD; Hangzhou, China)), then, using SPSS ((SPSS version 26.0, SPSS is a statistical software developed by three students of Stanford University in 1968; Palo Alto, CA, USA)) binary logistic regression analysis was performed and an early diagnosis model was constructed at 95% CI using the Wilcoxon test. ROC curves were used to analyze the predictive power of the model.

### 2.14. Determination of Core Genes

First we used the R packages “limma”, “GSEABase”, “GSVA”, and “ggplot2” to perform GSVA enrichment for the genes in the risk score and mode. The Spearman correlation test was used to analyze their correlation with the enriched pathways. According to the results of multivariate Cox proportional hazard regression, the genes that can be used as independent prognosis were screened, with *p* < 0.05 and HR > 1 as the screening criteria. Then, according to the ROC curve analysis of the gene in the early diagnosis model, the gene with the best diagnostic performance was screened. Finally, the core gene was determined according to the above three methods.

### 2.15. Correlation Analysis of Core Gene Expression and TME

R packages “limma” and “estimate” were used to score stromal cells and immune cells, and estimated respectively by CIBERSORT algorithm. The CIBERSORT algorithm analyzes gene expression data by performing 1000 permutations. The R packages “reshape2”, “limma”, and “ggpubr” were then used to evaluate their association with core gene expression in tumor tissues of GC patients, with *p* < 0.05 as the screening criterion.

### 2.16. Correlation Analysis of Core Gene Expression and Immune Cells and Immune Cell Infiltration

The R packages “preprocessCore” and “limma” were used to ssGSEA through the preprocessCore algorithm, which calculates a single sample. The scores of immune cell infiltration were evaluated using the R packages”limma”, “vioplot”, and “ggExtra” to evaluate the relationship between immune cell infiltration and core gene expression and draw a boxplot. Then the R packages “limma”, “corrplot”, and “ggpubr” were analyzed by the Spearman correlation test to analyze the correlation between immune cells and the expression of core genes in tumor tissues of GC patients, with *p* < 0.05 as the screening standard.

### 2.17. Analysis of the Correlation between Core Gene Expression and Immune Checkpoints and ICIs

The R packages “ggpubr”, “ggplot2”, and “corrplot” were used to analyze the correlation between the immune checkpoint and the expression of core genes in the tumor tissues of GC patients by the Spearman correlation test and draw the correlation heatmap, with *p* < 0.001 as the screening standard. The relationship between the expression of core genes in tumor tissues of GC patients and TCIA was also explored, and the expression matrix of TCGA GC and clinically relevant data of anti-PD1 and anti-CTLA4 blockers were downloaded from the TCIA online website. Using the R package “limma”, we evaluated the connection between CTLA4_negative + PD-1_positive and CTLA4_positive + PD-1_negative with CTLA4 _positive + PD-1_positive and the core genes, and then used the R package “ggpubr” to draw a violin plot, with *p* < 0.05 as the screening criterion.

## 3. Results

### 3.1. Identification of NRG Differentially Expressed in GC

The flow chart of this paper is shown in Figure 1. 381 patients were included in this study. A total of 740 NRGs were associated with GC, of which a total of 83 NRGs were differentially expressed between GC tumor and non-tumor tissues, with 36 differentially up-regulated and 47 differentially down-regulated (Figure 2A,B).

### 3.2. Gene Mutation and CNV Analysis of DE-NRGs

The gene mutation of DE-NRGs in GC is shown in Figure 3A,B. Genetic mutations of DE-NRGs were found in 68.2% (148/217) of GC samples (Figure 3A). Among these genes, as shown in Figure 3B, missense mutations were the most common variant classification, SNP were the most common variant typing, and C > T was the highest SNV category, with a maximum of 29 DE-NRGs mutated in each sample. FLNC is the gene with the highest mutation frequency, this was followed by MAP1B and SVIL (Figure 3A,B). In the CNV analysis, most DE-NRGs had copy number amplifications and deletions, with ADAMTSL4, TP63, and AHSG having widespread CNV amplifications and CDKN2A, EZH2, and NAT2 having widespread CNV deletions (Figure 3C). Figure 3D shows the location of CNV changes in DE-NRGs on chromosomes, with CNV changes mainly concentrated on chromosomes 1, 2, 4, 7, and 9.

### 3.3. GO and KEGG Analysis of DE-NRGs

As shown in Figure 4A,B, in the GO enrichment analysis, the biological processes of DE-NRGs mainly included cellular responses to biological stimuli, cellular responses to lipopolysaccharides, and cellular responses to bacteria-derived molecules. Furthermore, as shown in Figure 4C,D, KEGG enrichment indicated that DE-NRGs were mainly involved in the following pathways: (1) influenza A; (2) necroptosis pathway; (3) NOD-like receptor signaling pathway; and (4) the interaction pathway of viral proteins with cytokines and cytokine receptors.

### 3.4. Building a Prognostic Risk Scoring Model

The TCGA dataset (381) was randomly divided into Train (229) and Test (152) groups. The results of univariate Cox proportional hazards regression and LASSO regression analysis in the Train group showed that CCT6A, ZFP36, TP53I3, FAP, and BGN were associated with the prognosis of GC patients (Figure 5A–C). Multivariate Cox proportional hazards regression analysis showed that CCT6A, ZFP36, TP53I3, and FAP had independent prognostic potential (Figure 5D). The prognostic risk score model of these four genes was constructed as follows:risk score = (0.001742795 × expression of ZFP36) + (0.007973749 × expression of CCT6A) + (0.050876025 × expression of TP53I3) + (0.138885692 × expression of FAP).

The results of GEPIA showed (Figure 5E) that the expression levels of CCT6A and FAP were increased in the tumor tissues of GC patients, and the expression levels of ZFP36 and TP53I3 were decreased in the tumor tissues of GC patients. HPA shows that CCT6A, ZFP36, TP53I3, and FAP were all unfavorable prognostic factors (Figure 5F, Table 2). A risk score was calculated for each GC patient based on the expression levels of the 4 genes, and the patients were divided into high-risk and low-risk groups based on the median risk score (0.875). Figure 5G shows the OS of patients in the Train group, with a higher OS rate in low-risk GC patients than in high-risk GC patients. Figure 5H shows the ROC curve areas of 0.677, 0.759, and 0.866 at 1, 3, and 5 years in the Train group, respectively. Figure 5I shows that the results of PCA indicate that the risk score can appropriately separate the two risk groups.

### 3.5. Validation of Risk Score

Risk scores were respectively calculated for each GC patient in the Test group and the TCGA repertoire, based on the expression levels of the four genes. Figure 6A,B show that in the OS of Test and TCGA patients, the OS rate of GC patients in the high-risk group was lower than that of GC patients in the low-risk group. Figure 6C shows that the ROC curve area of the Test group at 1, 3, and 5 years is 0.613, 0.721, and 0.81, respectively; Figure 6D shows that the ROC curve area of TCGA at 1, 3, and 5 years is 0.647, 0.749, and 0.837, respectively. Figure 6E,F show that the results of the PCA indicate that the risk score can appropriately separate the two risk groups in the Test and TCGA sets. These results suggest that the risk score is a good predictor.

### 3.6. Independent Prognostic Assessment of Risk Score

Univariate Cox proportional hazard regression analysis revealed risk score (HR = 1.555, 95%CI: 1.336−1.810), stage (HR = 1.469, 95%CI: 1.095−1.972), and age (HR = 1.033, 95%CI: 1.010−1.057) and was a significant predictor of prognosis (Figure 7A). Multivariate Cox proportional hazards regression analysis showed that the risk score (HR = 1.584, 95%CI: 1.356−1.851) and age (HR = 1.046, 95%CI: 1.020−1.073) were independent predictors of outcome (Figure 7B). Figure 7C represents the C-index analysis results which show that the independent prediction of risk score is better than that of other clinicopathological features. Figure 7D shows that risk scores and gene expression in the model were related to clinicopathological features.

### 3.7. Nomogram

The nomogram is shown in Figure 8A. In addition, a patient’s risk score and clinicopathologic characteristics were marked on the chart to form a total score (409) and predict survival at 1, 3, and 5 years (1-year survival rate was 0.86; the 3-year survival rate was 0.65; the 5-year survival rate was 0.534) (Figure 8A). The calibration curve, shown in Figure 8B, demonstrates the stability of the nomogram. The ROC curve area showed that the risk score and nomogram predictive power were superior compared with other clinicopathological features (Figure 8C). The DCA curve shows better predictive ability of the risk score model and nomogram (Figure 8D).

### 3.8. GSEA

Based on the analysis of the GSEA results, the KEGG results (Figure 9A) showed that many immune and inflammatory pathways were activated in the high-risk group, such as cytokine and cytokine receptor pathways, ECM receptor interactions, epithelial cell signaling pathways in H. pylori infection, and TOLL-like receptor signaling pathways; there were no significantly enriched pathways in the low-risk group. The GO results (Figure 9B) show that the biological processes of collagen metabolism, negative regulation of inflammatory response, and positive regulation of sodium ion transmembrane transport were significantly enhanced in the high-risk group. The positive regulation of sodium ion transmembrane transporter activity, the negative regulation of food antibody response, and the production of transforming growth factor beta biological processes were mainly concentrated in the low-risk group.

### 3.9. Identification of Different Risk Groups Associated with Immunity

The results showed that the ImmuneScore, StromalScore, and ESTIMATEScore were lower in the low-risk group than in the high-risk group in TME (Figure 9C), and the ImmuneScore (R = 0.17, *p* = 0.00068), StromalScore (R = 0.34, *p* = 9.2 × 10^-12^), and ESTIMATEScore (R = 0.28, *p* = 3.4 × 10^-8^) were positively correlated with the risk score (Figure 9D–F). In the infiltration of immune cells, SSGSEA results show that the infiltration scores of aDC, TIL cells, iDCs, PDC cells, Th1 cells, Th2 cells, CD8 + T cells, DCs, macrophages, Neutrophils, NK cells, Treg, and T helper cells in the high-risk group were higher than those in the low-risk group (all *p* < 0.05). However, there was no significant difference in infiltration scores of B cells and Tfh cells between the two risk groups (Figure 9G). There was no significant difference in the infiltration scores of MHC class I and type II interferon response function between the two risk groups (*p* > 0.05). In the high-risk group, the infiltration scores of CCR, HLA, APC co-inhibition, APC co-stimulation, T cell co-inhibition, T cell co-stimulation, checkpoint, inflammation-related promotion, para-inflammation, cytolytic activity, and type I interferon response function were higher than those in the low-risk group (Figure 9H). Figure 9I shows that cancer-associated fibroblasts (CAF) have the strongest correlation with risk scores. In order to study the potential changes between immune checkpoint expression in high and low risk groups, a total of 47 immune checkpoint-related genes were included, among which CD200, CD244, TNFSF9, CD274, KIR3DL1, PDCD1LG2, CD86, CD28, TIGIT, NRP1, TNFRSF4, CD70, TNFRSF18, HAVCR2, LAIR1, CD80, TNFSF4, CD40, TNFSF18, IDO1, TNFRSF9, CTLA4, LAG3, ICOS, CD276, CD48, and TNFRSF8 were more expressed in high-risk groups than in low-risk groups (Figure 9J).

### 3.10. The Importance of Risk Scores in Immunotherapy Response and ICIs

The results showed that the Dysfunction score, Exclusion score, CAF score, and TIDE score of the low-risk group were lower than those of the high-risk group (Figure 10A–D). The results also showed that the use of anti-CTLA4 blockers varied in the high and low-risk groups (Figure 10E). The use of anti-PD1 blockers and the combination of anti-CTILA4 blockers with anti-PD1 blockers were not statistically significant in the high and low-risk groups (Figure 10F,G).

### 3.11. Identification of 4-Gene Drug Susceptibility and Resistance

The results of drug sensitivity and drug resistance analysis showed that the expression of ZFP36 in tumor tissues of GC patients was associated with resistance to CGP60474 and gefitinib, and sensitivity to camptothecin, SB52334, talazopanib, and TW37. The expression of TP53I3 in GC was not correlated with drug sensitivity and drug resistance. The expression of FAP in GC was correlated with the sensitivity of TGX221 and TW37. CCT6A expression in GC tumor tissues was associated with resistance to CGP60474 and SB52334, and sensitivity to camptothecin, TGX221, talazopanib, etoposide, gemcitabine, HG511301, TW37, vinorelbine, and gefitinib (Figure 10H).

### 3.12. Construction and Identification of Early Diagnostic Models

Analysis of results showed that (Figure 11A–D) the expression of ZFP36 (normal vs. Stage I: *p* = 0.0138013; normal vs. stage II: *p* = 0.031858), FAP (normal vs. stage I: *p* = 0.049636; normal vs. stage II: *p* < 0.001), and CCT6A (normal vs. stage I: *p* < 0.001; normal vs. stage II: *p* < 0.001) in normal tissues of GC patients and tumor tissues of GC patients with clinical pathology stage 1 and stage 2 were statistically significant (both *p* < 0.05). The expression of TP53I3 (normal and stage I: *p* = 0.58422; normal and stage II: *p* = 0.2748) was not statistically significant (*p* > 0.05). The results showed that ZFP36, FAP, and CCT6A could be used as markers for the early diagnosis of GC patients. The expression matrix of GSE26899 normal tissues and tumor tissues of GC patients with clinical pathology of stage I and stage II (12 normal samples and 29 tumor samples) was used. Binary logistic regression analysis showed that FAP and CCT6A could be used as markers for early diagnosis, and their early diagnosis model was constructed. The formula of the model was as follows:early diagnosis score = −64.626 + 5.059 × CCT6A expression + 2.963 × FAP expression.

The true positive rate of the model was 93.1%, the true negative rate was 83.3%, and the accuracy was 90.2% (Table 3). The combined diagnostic efficacy was AUC = 0.943 (95% CI: 0.873–1.000) (Figure 11E).

### 3.13. Identification of Core Gene

The analysis of relevant results showed that, according to the analysis of GSVA enrichment results (Figure 11F), the main enriched pathways of FAP were more similar to the risk score compared with other genes. ZFP36, TP53I3, CCT6A, and FAP were shown to be independent prognostic factors in multivariate Cox proportional hazards regression (all *p* < 0.05) (Figure 5D). In that binary logistic regression analysis (Figure 11E), the diagnostic efficacy of CCT6A was AUC = 0.776, and that of FAP was AUC = 0.922. According to the efficacy of early diagnosis, the diagnostic efficacy of FAP was better than that of CCT6A. In conclusion, FAP can be used as a core gene.

### 3.14. Comprehensive Analysis of Core Gene Expression

According to the results of GSVA enrichment (Figure 11F), FAP was mainly enriched in TGFβ signaling pathway, JAK-STAT signaling pathway, and MAPK signaling pathway. Figure 12A shows that FAP was statistically significant with StromalScore, ImmuneScore, and ESTIMATEScore. Additionally, Figure 12B shows that the expression of FAP in tumor tissues of GC patients was significantly different from that of primitive B cells, B memory cells, plasma cells, regulatory T cells, T gamma delta cells, and macrophage M2 among immune cells. It is also related to macrophage M0, regulatory T cells, macrophage M2, eosinophils, primitive B cells, B memory cells, and plasma cells. There was a positive correlation with macrophage M0, regulatory T cells, macrophage M2, and eosinophils, and a negative correlation with blast B cells, B memory cells, and plasma cells (Figure 12C). The expression of FAP in the tumor tissues of GC patients was strongly correlated with CD200, PDCD1LG2, CD86, NRP1, HAVCR2, LAIR1, TNFSF4, and CD276 among the genes related to immune checkpoints, mainly being positively correlated (Figure 12D,E). Analysis of FAP expression in the tumor tissues of GC patients and TCIA score showed (Figure 12F–H) that in CTLA4_negative + PD-1_positive, CTLA4_positive + PD-1 _negative, and CTLA4_positive + PD-1_positive, the low expression level of FAP in GC tumor tissues was significantly lower than the high expression level (all *p* < 0.01).

## 4. Discussion

Necroptosis is a form of programmed cell death, which plays an important role in the initiation and progression of many cancers [8,9]. RIPK3 and MLKL are the core components of necroptosis. It has been reported that the expression of MLKL is closely related to the prognosis of GC patients [14], which indicates that necroptosis is closely related to GC. However, the potential molecular mechanism of NRG in GC has not been reported. We believe that, besides this key gene, other NRGs may play a more powerful role in the occurrence and development of GC. It is of great significance to explore the role of more NRGs in the occurrence and development of GC. It would be of great practical value to construct early diagnostic and prognostic features composed of NRGs. Therefore, in this study, we evaluated the relationship between early diagnosis and prognosis of GC patients and NRGs, and identified biomarkers and therapeutic targets of NRGs associated with GC.

First, in our study, a prognostic risk score model for 4-NRGs (TP53I3, ZFP36, FAP, and CCT6A) was established by multivariate Cox proportional hazards regression. The model classifies GC patients into high-risk and low-risk groups, mainly by the median value of the risk score. Patients in different risk groups showed different OS, and the results showed that GC patients in the high-risk group had a relatively shorter survival time than those in the low-risk group. The GEPIA database was used to verify that ZFP36 and TP53I3 were down-regulated in GC tissues, and FAP and CCT6A were up-regulated in GC tissues. The HPA database also analyzed the protein expression of these four NRGs in tumor tissues and normal tissues; these results indicated that all 4 genes are pro-cancer factors. Our research showed that the prognostic risk score model consisting of ZFP36, TP53I3, FAP, and CCT6A proved to be highly predictive. We then demonstrated that the NRG prognostic risk score was an independent prognostic factor based on multivariate Cox proportional hazards regression. Next, we constructed a nomogram for the NRG prognostic risk score model and used it to accurately predict survival at 1, 3, and 5 years in patients with GC. ROC curve and DCA decision curve were used to evaluate the predictive ability of the nomogram, and showed that the NRG prognostic risk score and nomogram had good predictive effects compared with other clinicopathological features. In addition, we divided the TCGA complete set into Train and Test groups in different proportions, and confirmed that the predictive power of the models grouped by 6:4 was better than the other proportions (Appendix A).

We also explored the enrichment and characteristics of different risk groups using GSEA. It was found that the high-risk group was mainly enriched in the inflammatory and immune pathways, which means that necrotizing apoptosis is closely related to inflammation and immunity, and these pathways also play a huge role in the occurrence and development of GC.

In recent years, immunotherapy has achieved great results in GC patients [28,29,30,31], so we analyzed the correlation between risk score and immunity. Our study shows that the NRG prognostic risk score has important associations with TME, immune cells, immune cell infiltration, immune checkpoints, and immunotherapy. Among them, TME is an important biological environment for the occurrence and development of tumors, and is also closely related to immunotherapy. The main components are stromal cells and immune cells. GC is a typical inflammation-related malignancy, and its microenvironment contains a large number of immune cells. Our study demonstrates that different risk groups in the NRG prognostic risk score are associated with stromal cells and immune cells in the immune microenvironment. In the high-risk group, their content was also higher when compared with the low-risk group. The role of immune cells in infiltrating the tumor microenvironment and immune checkpoint genes in different risk groups of GC patients was also investigated, highlighting potential therapeutic targets for different risk groups in GC patients. Among them, we found that in the ssGSEA analysis, the infiltration fraction of most immune cells was lower in the low-risk group than in the high-risk group, such as macrophages, neutrophils, NK cells, Treg, and T helper cells. However, in terms of immune checkpoints, most of the related genes are highly expressed in high-risk groups, such as CTLA4 and PD-L2. Our study also shows that immunotherapy is effective between different risk groups.

We then examined the association between the 4 genes and early diagnosis in the prognostic risk score model. Our study shows that FAP, ZFP36, and CCT6A are related to the early diagnosis of GC. We then constructed a 2-gene (FAP and CCT6A) early diagnosis model using binary logistics regression. In addition, we also constructed a 3-gene early diagnosis model, but the 3-gene early diagnosis model had a lower accuracy rate (87.8%) compared with the 2-gene diagnosis model (90.2%) (Appendix A). Therefore, we determined that the 2-gene early diagnosis model was better than the other diagnostic models.

Finally, we identified FAP as the core gene. We then performed a series of analyses on FAP. FAP has been reported to be a type II membrane-bound glycoprotein; it is one of a family of serine proteases, which are usually proteins expressed in fibroblasts, such as CAF, which is a major component of TME [32]. When FAP is overexpressed in GC patients, it can increase the malignancy of the tumor, promote the proliferation, migration, invasion, and apoptosis inhibition of SGC7901 cells, and induce apoptosis of GES1 cells in vitro [33]. FAP can also induce the drug resistance of cell lines, thereby inhibiting the anti-tumor ability of T cells in GC TME, and can also promote the anti-tumor effect of ICIs in GC patients [19]. Other studies have shown that the high expression of FAP is positively correlated with the angiogenesis and metastasis of GC [34]. In addition, FAP is closely related to the prognosis of lung adenocarcinoma, ovarian cancer, and colorectal cancer [35,36,37]. However, no relevant study has confirmed that FAP plays a certain role in the early diagnosis of gastric cancer. Our study showed that FAP is an independent prognostic factor and also a biomarker for early diagnosis, with an individual diagnostic efficacy of AUC = 0.922. Furthermore, our study showed that FAP is mainly enriched in some immune-related pathways, such as TGFβ signaling pathway, JAK-STat signaling pathway, and MAPK signaling pathway. The expression of FAP in the tumor tissues of GC patients is different in StromalScore, ImmuneScore, and ESTIMATEScore, and the higher the FAP expression in tumor tissues of GC patients, the higher the TME score. The expression of FAP in the tumor tissues of GC patients was also related to most immune cells and was positively correlated, such as regulatory T cells, macrophages M2, and eosinophils. It is also positively and strongly correlated with most immune checkpoints, such as CD200 and PDCD1LG2. However, ICIs also play an important role in GC. Our study showed that FAP expression levels in tumor tissues were elevated in GC patients. It may be possible to benefit from treatment with anti-PD-1 receptor or anti-CTLA4 receptor alone or a combination of anti-PD-1 and anti-CTLA4 receptor blockers.

However, there are some limitations in our study. First, the data used were from retrospective studies in the TCGA database and the GEO database and were not prospective. Prognostic risk score models and early diagnosis models of GC associated with NRG were then established and were not validated by clinical organizations. As well as our use of preclinical models to reveal that NRG may be a prognostic biomarker, and the analysis of risk scores in GC patients in the TME, immune cells, and ICIs studies, we need to use relevant samples to respectively verify them at the mRNA and protein levels, and support our findings. The study of the core gene also needs to be further verified at the protein level.

## 5. Conclusions

In summary, we conducted a comprehensive bioinformatics study of NRGs in GC, and determined the early diagnosis and prognosis characteristics of NRGs in GC. Using machine learning, we established a 2-NRG early diagnosis model to predict the occurrence of early GC and a 4-NRG prognosis risk assessment model to evaluate the prognosis of GC patients, both of which have good predictive ability. Then, the risk score and TME, immune cell infiltration, immune checkpoint, ICIs, and so on were analyzed in depth, which can further prove that our prognostic risk assessment model has high practical value. The core gene, FAP, which can be used as both an early diagnostic marker and an independent prognostic factor, was also identified. In addition, our study also revealed the relationship between ICIs and different expressions of FAP in GC tumor tissues, which provides a new target for immunotherapy of GC, and provides new ideas and theoretical basis for basic research, clinical diagnosis, and individualized treatment of GC.

## Figures and Tables

**Figure 1 cancers-14-03891-f001:**
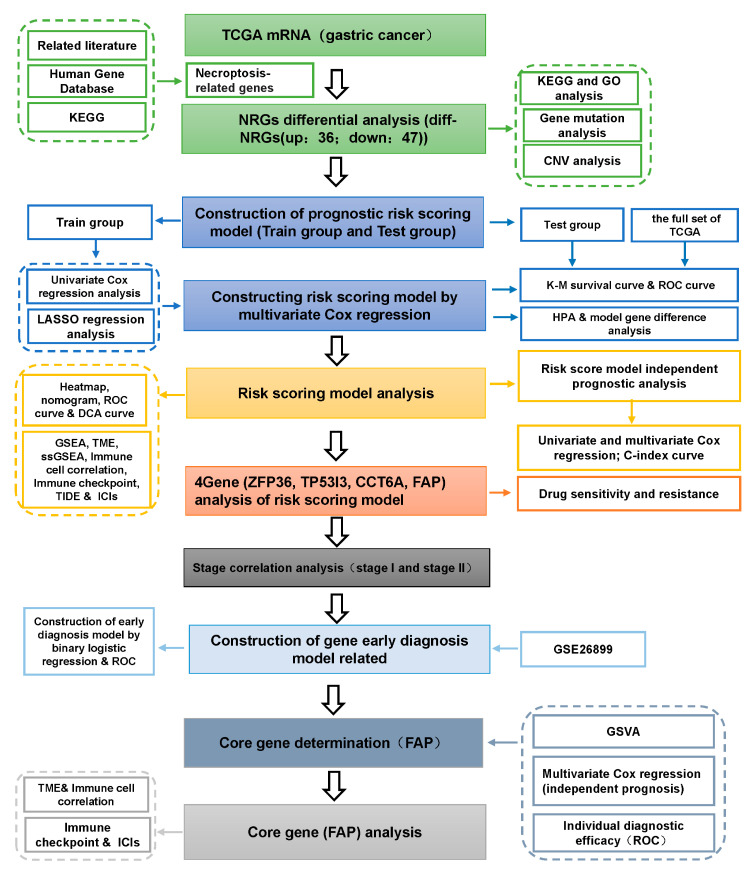
Flow chart. (NRGs: necroptosis-related genes; ICIs: immune checkpoint inhibitors; TME: mmune microenvironment; TCGA: the Cancer Genome Atlas; KEGG: Kyoto Encyclopedia of Genes and Genomes; Diff-NRGs: differentially expressed necroptosis-related genes; GO: Gene Ontology; CNV: copy number variation; LASSO: absolute minimum shrinkage and selection operator; K-M: Kaplan-Meier; ROC: receiver operating characteristic; HPA: Human Protein Atlas; C-index: concordance index; ssGSEA: single-sample enrichment analysis; TIDE: Tumor Immune Dysfunction and Exclusion.

**Figure 2 cancers-14-03891-f002:**
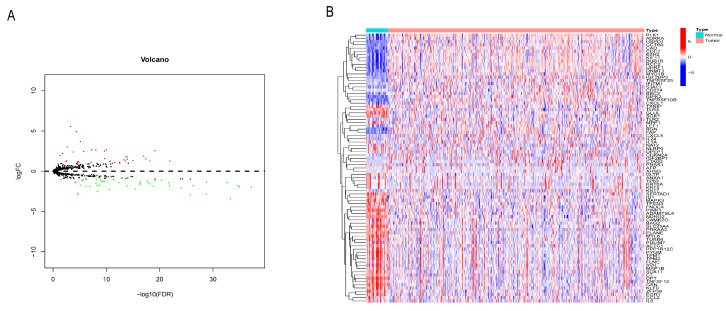
Expression analysis of NRGs in GC. (**A**) Volcano map (red dots represent up-regulated genes, green dots represent down-regulated genes and black dots represent the genes with no difference). (**B**) diff-NRGs heatmap (red represents up-regulated genes, blue represents down-regulated genes).

**Figure 3 cancers-14-03891-f003:**
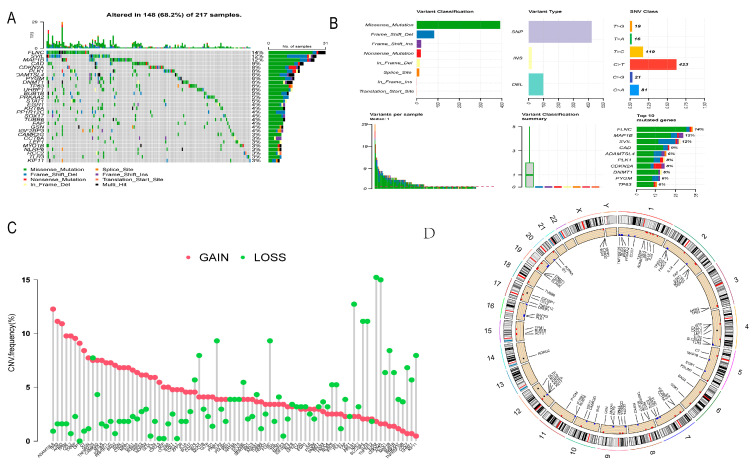
Mutation and CNV analysis of NRGs in GC. (**A**,**B**) The mutation frequency and classification of diff-NRGs in GC. (**C**) Copy number deletion and amplification of diff-NRGs (red for GAIN, green for LOSS). (**D**) CNV changes of diff-NRGs on chromosomes.

**Figure 4 cancers-14-03891-f004:**
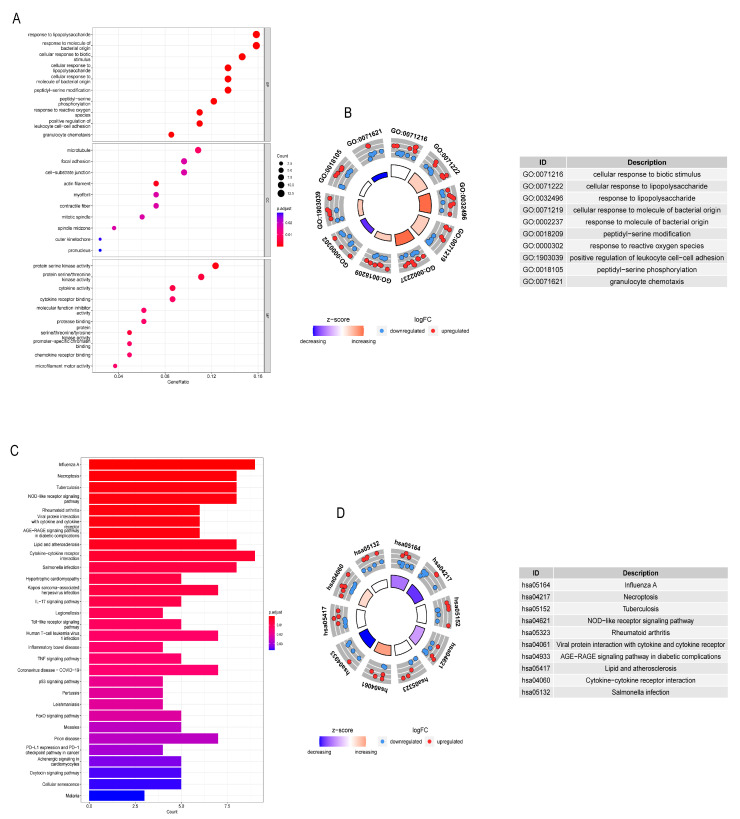
GO and KEGG analysis of NRGs in GC. (**A**,**B**) GO analysis of diff-NRGs. (**C**,**D**) KEGG analysis of diff NRGs. (all *p* < 0.05).

**Figure 5 cancers-14-03891-f005:**
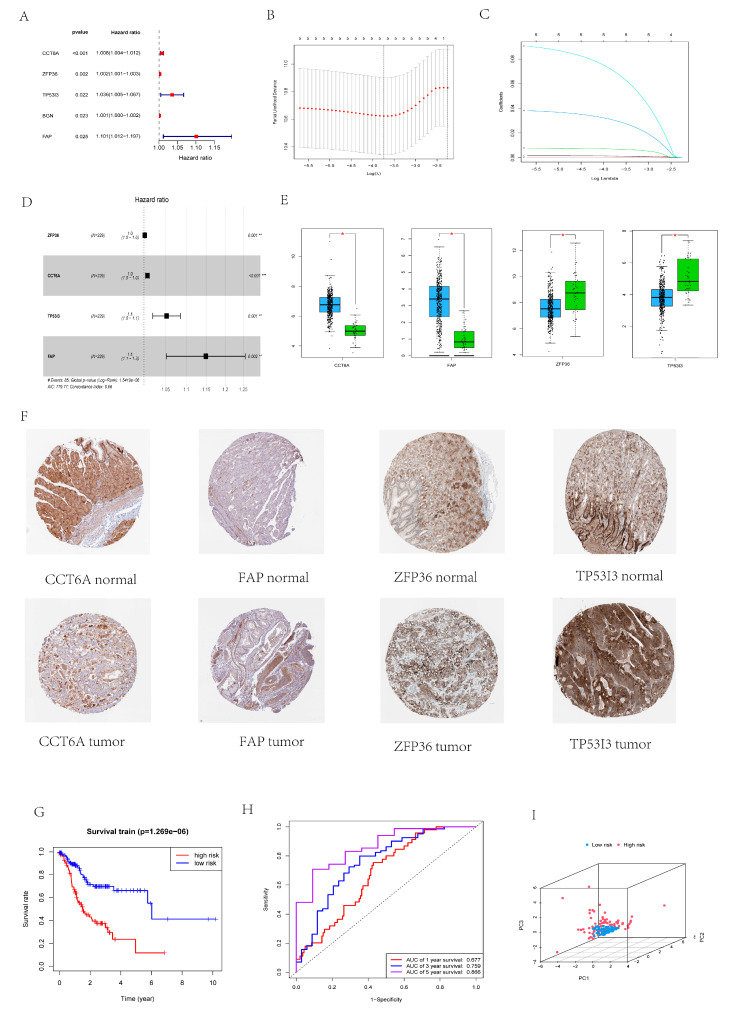
Construction of a necroptosis-related prognostic gene signature**.** (**A**) Forest map showed 5 necroptosis-related prognostic regulators for GC identified by univariate Cox proportional hazard regression analysis. (**B**,**C**) The coefficient and partial likelihood deviance of prognostic signature. (**D**) The forest map showed four independent prognostic regulators related to necrotic apoptosis in GC determined by multivariate proportional hazard regression analysis (**E**) Expression of four independent prognostic factors related to necrotic apoptosis in GC (blue represents tumor tissue and green represents normal tissue). (**F**) Expression of four independent prognostic factors related to necrotic apoptosis in HPA. (**G**) OS curve of Train group (red represents high-risk population and blue represents low-risk population). (**H**) ROC curve analysis of Train group. (**I**) PCA analysis of Train group. (* *p* < 0.05, ** *p* < 0.01, *** *p* < 0.001).

**Figure 6 cancers-14-03891-f006:**
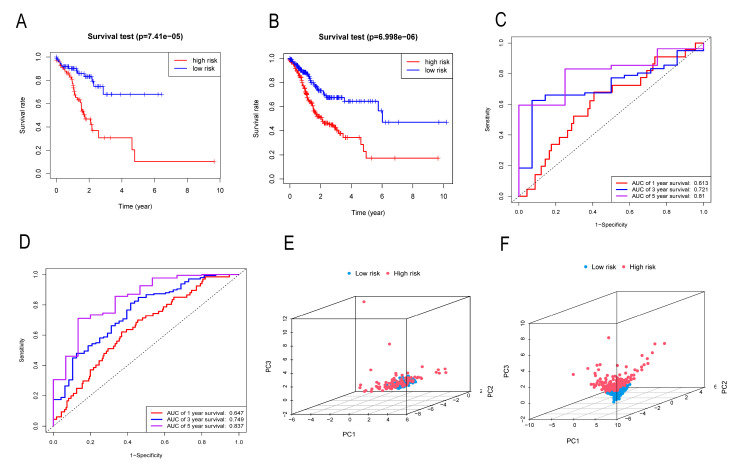
Validation of risk score. (**A**,**B**) Validation of the OS curve of Test group and TCGA complete set. (**C**,**D**) Test group and TCGA complete set ROC curve verification. (**E**,**F**) PCA verification of Test group and TCGA complete set.

**Figure 7 cancers-14-03891-f007:**
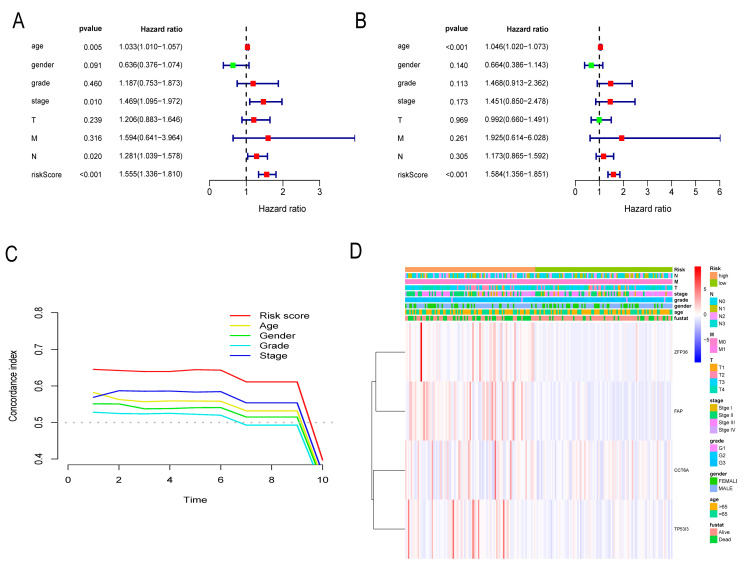
Independent prognosis and clinical-related identification. (**A**) Univariate risk proportional regression showed that risk score was significantly correlated with OS. (**B**) Multivariate proportional hazard regression showed that the risk score was an independent prognostic indicator of poor GC outcomes. (**C**) The c-index curve showed that the independent prognostic ability of risk score was better than that of other clinical features. (**D**) Heatmap of risk score, model gene, and clinical characteristics.

**Figure 8 cancers-14-03891-f008:**
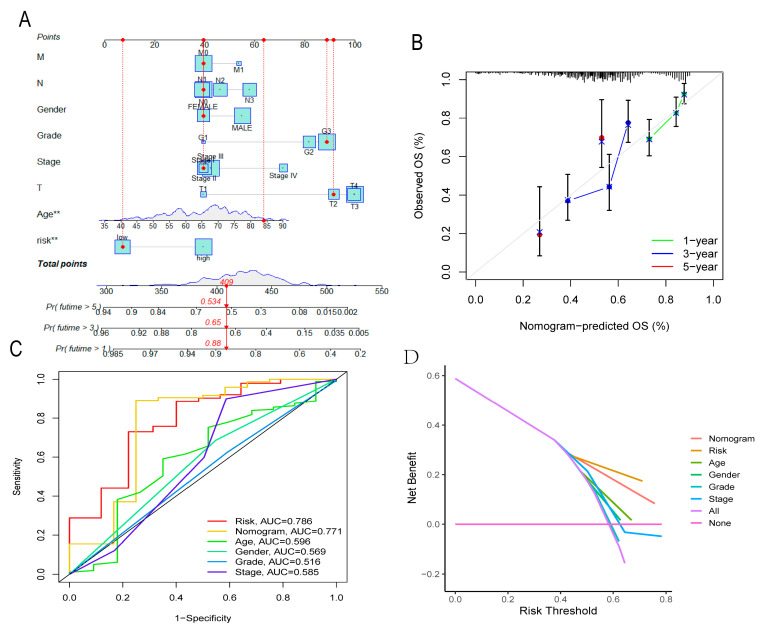
Nomogram and identification. (**A**) Nomogram. (**B**) The calibration curve determines the stability of the nomogram. (**C**,**D**) Multi-index ROC curve and DCA decision curve show better predictive ability of nomogram and risk score. (all *p* < 0.05).

**Figure 9 cancers-14-03891-f009:**
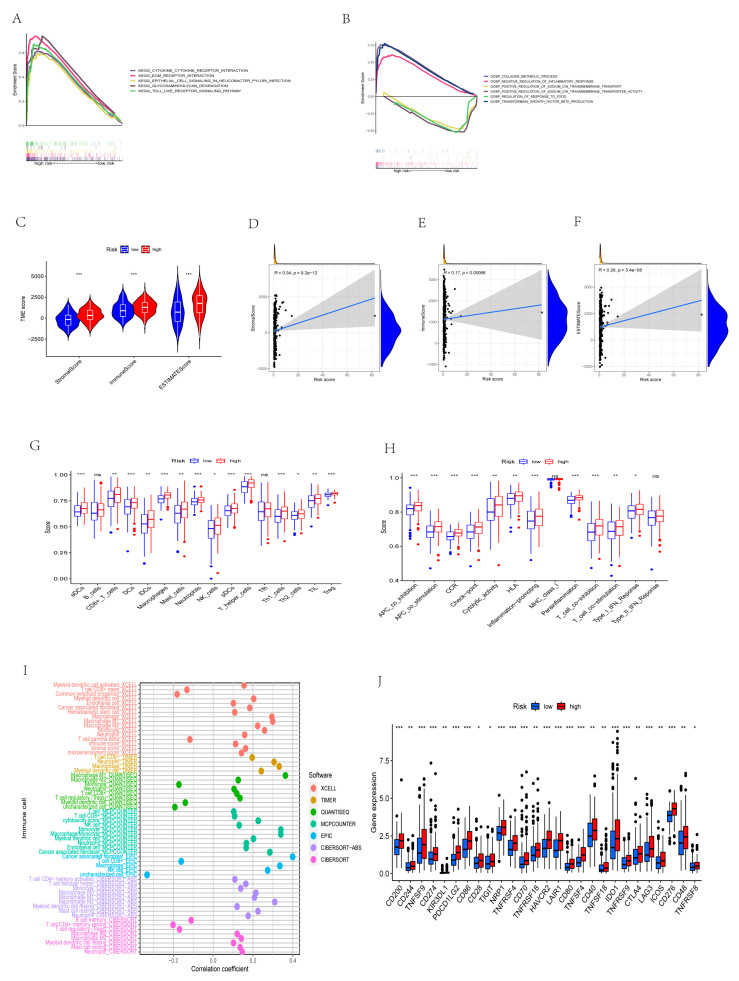
GSEA, TME, immune cell infiltration, and immune checkpoint identification of risk score. (**A**) KEGG enrichment in high-risk populations. (**B**) GO enrichment of different risk groups. (**C**) The situation of different risk groups in TME. (**D**–**F**) Correlation analysis between risk score and StromalScore, ImmuneScore, and ESTIMATEScore. (**G**) Expression of immune cells in different risk groups (ns represent nonsense). (**H**) Different cell functions in different risk groups (ns represent nonsense). (**I**) Correlation analysis of immune cells and risk scores in different algorithms. (**J**) Expression of immune checkpoints in different risk groups. (* *p* < 0.05, ** *p* < 0.01, *** *p* < 0.001).

**Figure 10 cancers-14-03891-f010:**
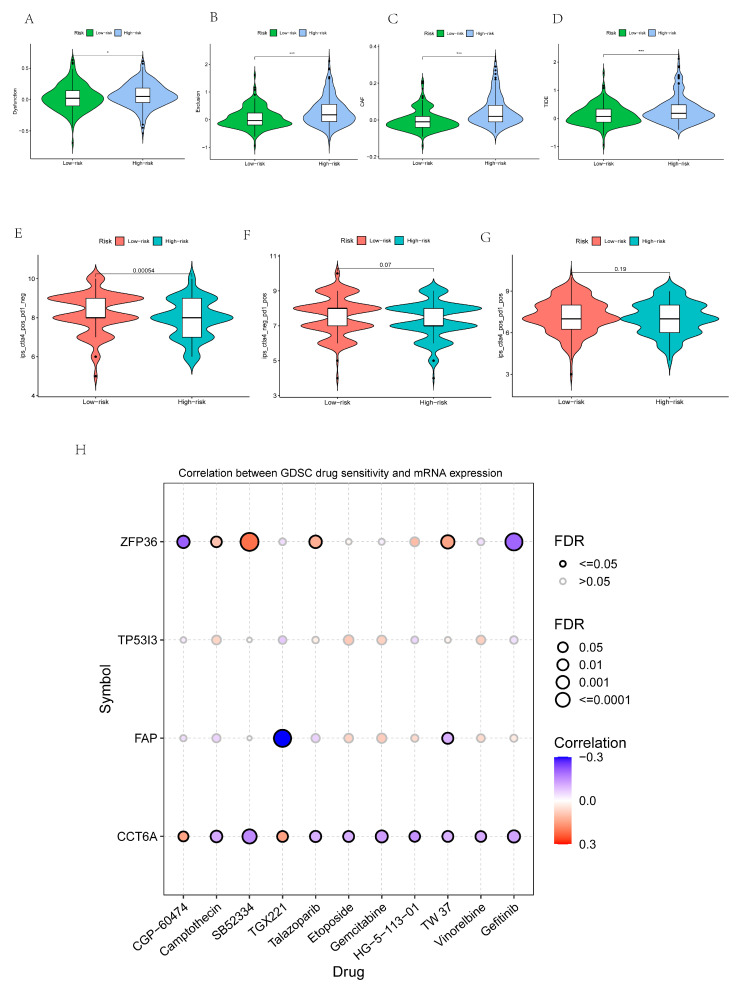
Identification of risk score and immunotherapy, ICIs and drug sensitivity and drug resistance of model genes. (**A**–**D**) analysis of Dysfunction, Exclusion, CAF and TIDE with risk score. (**E**–**G**) Anti-CTLA4 blockers, anti-pd1 blockers and the combination of anti-CTLA4 blockers with anti-pd1 blockers at different risks. (**H**) Analysis of drug sensitivity and drug resistance of four independent prognostic regulators related to necrotic apoptosis. (* *p* < 0.05, *** *p*< 0.001).

**Figure 11 cancers-14-03891-f011:**
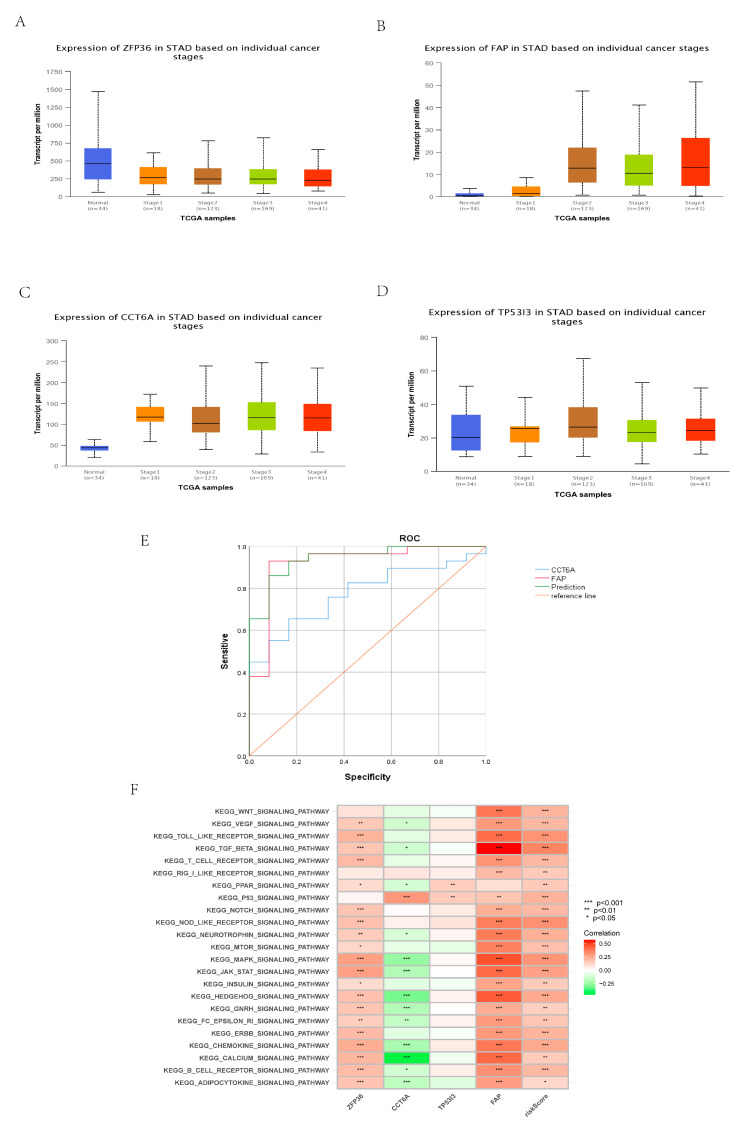
Early diagnosis and identification of GSVA. (**A**–**D**) Analysis of the expression and normal expression of four independent prognostic regulators related to necroptosis in different stages of GC. (**E**) ROC verifies the predictive ability of the early diagnosis model. (**F**) GSVA. (* *p* < 0.05, ** *p* < 0.01, *** *p* < 0.001).

**Figure 12 cancers-14-03891-f012:**
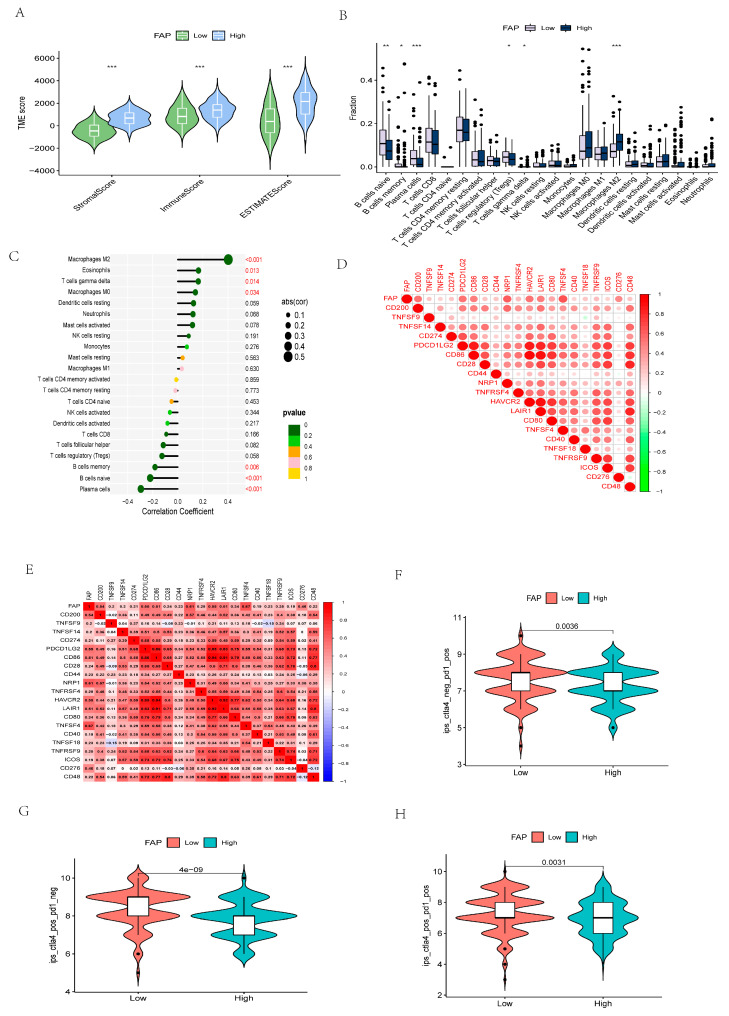
Related identification of core gene. (**A**) The expression of core gene and the analysis of TME. (**B**) Expression of core gene and analysis of immune cells. (**C**) Correlation analysis between immune cells and core gene. (**D**,**E**) Correlation analysis between core gene and immune checkpoint related genes. (**F**–**H**) Analysis of different expressions of anti-CTLA4 blockers, anti-pd1 blockers, and the combination of anti-CTLA4 blockers with anti-pd1 blockers in core gene. (* *p* < 0.05, ** *p* < 0.01, *** *p* < 0.001).

**Table 1 cancers-14-03891-t001:** Clinical characteristics.

TCGA		GEO	
Variables	Number of Samples	Variables	Number of Samples
Gender		Gender	
Male	159	Male	75
Female	290	Female	21
		Unknown	12
Age at diagnosis		Age at diagnosis	
≤65	198	≤65	63
>65	242	>65	33
Unknown	9	Unknown	12
Grade		Grade	
G1	12	G1	Unknown
G2	159	G2	Unknown
G3	263	G3	Unknown
G4	9	G4	Unknown
Unknown	6	NA	Unknown
Stage		Stage	
I	59	I	11
II	131	II	18
III	184	III	27
IV	44	IV	36
Unknown	31	NA	16
T		T	
T1	23	T1	Unknown
T2	93	T2	Unknown
T3	200	T3	Unknown
T4	119	T4	Unknown
Unknown	14	NA	Unknown
M		M	
M0	393	M0	Unknown
M1	30	M1	Unknown
Unknown	26	NA	Unknown
N		N	
N0	133	N0	Unknown
N1	120	N1	Unknown
N2	85	N2	Unknown
N3	88	N3	Unknown
Unknown	23	NA	Unknown
fustat		fustat	
Alive	277	Alive	Unknown
Dead	172	Dead	Unknown

NA: not available.

**Table 2 cancers-14-03891-t002:** HPA Patient information.

Gene	Antibody Number	Patient Id	Type	Staining	Intensity	Quantity
ZFP36	HPA006009	2130	Normal tissue	Medium	Moderate	>75%
ZFP36	HPA006009	2195	Adenocarcinoma	High	Strong	>75%
TP53I3	HPA022012	2583	Normal tissue	Medium	Strong	<25%
TP53I3	HPA022012	3270	Adenocarcinoma	High	Strong	>75%
FAP	HPA059739	1924	Normal tissue	High	Strong	75–25%
FAP	HPA059739	2378	Adenocarcinoma	High	Strong	>75%
CCT6A	HPA042996	3379	Normal tissue	High	Strong	>75%
CCT6A	HPA042996	2105	Adenocarcinoma	High	Strong	>75%

**Table 3 cancers-14-03891-t003:** Accuracy of early diagnosis.

	GC	Normal	All	Accuracy
**2-mRNA positive**	27	2	29	93.1
**2-mRNA negative**	2	10	12	83.3
**All accuracy**	27	10	41	90.2

## Data Availability

The data presented in this study are available on request from the corresponding author.

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
