# Peer review of "Novel Necroptosis-Related Gene Signature for Predicting Early Diagnosis and Prognosis and Immunotherapy of Gastric Cancer"

_cancers, 2022, doi:10.3390/cancers14163891_

Round 1

Reviewer 1 Report

I read the paper by Zhou et al. the paper is very intriguing and tackles a very important and new topic in the oncology field.

However before pubblication minor revision are required:

Simple abstract: MLKL indicate the meaning

Table 1 should be improuved it is difficult to read

Figure 1 is difficult to read. It should improuved e.g by coulors

Results 3.9 line 392-411 this part shoul be better explain; in particular down ed up-regulation.

Discussion: data about immune infiltrate should be better discuted

Author Response

Dear reviewer:

Thank you for your comments on our submitted article entitled "Novel necroptosis-related gene signature for predicting early diagnosis and prognosis and simultaneous of gastric cancer" (cancers-1813198). These comments are of great value and are of great help to us in revising and improving our articles. In response to your comments, we have carefully studied the comments and made corrections, hoping to get your approval. The main revisions of the paper and the replies to reviewers are as follows:

  1. Simple abstract: MLKL indicate the meaning.

In view of this comment, we do have some shortcomings in this part, which can not be well reflected in the article, so we have made some modifications in the preface lines 55-71 and discussion lines 516-521 and explained our ideas here. The purpose of quoting MLKL is to show that necroptosis or GC is closely related, and perhaps more necroptosis-related genes can be used in the study of gastric cancer, which can also prove the feasibility of our study.

  1. Table 1 should be improuved it is difficult to read.

We very much agree with your suggestion. There is indeed dyslexia in this form. In response to this suggestion, we have made improvements in lines 109-110 of the main text.

  1. Figure 1 is difficult to read. It should improuved e.g by coulors.

We attach great importance to your suggestion. Figure 1 does have dyslexia. In response to this suggestion, we have carefully revised it in line 270 of the text as required.

  1. Results 3.9 line 392-411 this part shoul be better explain; in particular down ed up-regulation.

We very much agree with your suggestion, and we do have many shortcomings in this part. We have explained this suggestion in more detail based on our data in lines 407-424 of text results 3.9.

  1. Discussion: data about immune infiltrate should be better discuted.

We agree with your suggestion very much, and we really discuss this part too briefly. In response to this suggestion, we have discussed it in more detail in lines 564-566 of the text discussion.

This is our reply. We have tried our best to further improve our article and revised it strictly according to the opinions of reviewers. Thank you to Reviewer for your enthusiastic work, hope our correction will be recognized, and thank you again for giving us many valuable suggestions.

Kind regards,

Dr. Zhou

Contributors

E-Mail: [email protected]

Reviewer 2 Report

The introduction is deprived of the related work with the recent literature.

Figure.1 needs more detail and it will be better to add the abbreviations in the description of figure or footnote. 

Figure 2 A and 2B needs more details. What they represent?

Figures 3 A,B,D are too blurred.  Provide better quality image.

How authors have determined the core genes in section 2.4. 

What are the practical implications of your research? 

Authors should further clarify and elaborate novelty in their contribution.

Conclusion is too short. Add more explanation. 

What are the limitations of the present work?

Author Response

Dear reviewer:

Thank you for your comments on our submitted article entitled "Novel necroptosis-related gene signature for predicting early diagnosis and prognosis and simultaneous of gastric cancer" (cancers-1813198). These comments are of great value and are of great help to us in revising and improving our articles. In response to your comments, we have carefully studied the comments and made corrections, hoping to get your approval. The main revisions of the paper and the replies to reviewers are as follows:

  1. The introduction is deprived of the related work with the recent literature.

Thank you for your suggestion. We do have shortcomings here. In view of this suggestion, although we have added recent relevant literature in the preface, we did not explain it specifically, but only briefly mention it. To solve this problem, we have made relevant supplements in lines 73-76 of the preface.

  1. 1 needs more detail and it will be better to add the abbreviations in the description of figure or footnote.

We agree with your suggestion very much. There are many brief abbreviations in Figure 1. In view of this deficiency, and in order to make our diagram more directly understood by more people, we have modified Figure 1 and added abbreviations to the description of the diagram (See lines 270-279 of the text for details).

  1. Figure 2A and 2B needs more details. What they represent?

We very much agree with your suggestion, and some details in Figure 2 are indeed insufficient. In view of this deficiency, we correct it in more detail in lines 281-284 of the text.

  1. Figures 3 A,B,D are too blurred. Provide better quality image.

Thank you for this suggestion. The ambiguity of Figure 3 is our negligence. In view of this deficiency, we have further adjusted the clarity of Figure 3 and replaced it (See line 298 for details).

  1. How authors have determined the core genes in section 2.4.

Thank you for your suggestion. In response to this proposal, Our explanation is: In section 2.4, we analyzed the gene mutation, CNV variation, KEGG and GO of the necroptosis-related genes(NRGs) after the difference. The purpose is to explore whether the NRGs are related to the occurrence and development of gastric cancer, which is helpful for us to find a better research direction and prove the feasibility of our research (see lines 124-132 and 285-309 of the text for details). However, in Section 2.14, we aim to identify the core genes: we identify our core genes through three aspects. Firstly, we found that the pathway enriched by FAP is more strongly correlated than other genes by GSVA enrichment analysis, and it is positively correlated; Then, through multivariate cox proportional risk regression analysis, we found that FAP, CCT6A, ZFP36 and TP53I3 (P < 0.05) can all be used as independent biomarkers for prognosis. Finally, through ROC analysis in early diagnosis, FAP alone is more effective than CCT6A. Based on these three aspects, we found that FAP can be used as a biomarker for both independent prognosis and early diagnosis. Compared with other candidate genes, FAP has better performance than CCT6A, ZFP36 and TP53I3 in all aspects. Therefore, we finally determined that FAP is the core gene of this paper (See lines 229-237 and 474-482 for details).

  1. What are the practical implications of your research?

Thank you for your suggestion. In response to this suggestion, our explanation is that we study necroptosis-related genes (NRGs) by analyzing the public transcriptome data of gastric cancer samples by machine learning, and determine the characteristics of necroptosis-related early diagnosis and prognosis genes of gastric cancer(GC). 2-NRGs (CCT6A and FAP) and 4-NRGs (ZFP36, TP53I3, FAP and CCT6A) can effectively evaluate the risk of early GC and the prognosis of GC patients, respectively. Through further analysis, we found that 4-NRGs are significantly related to immunotherapy effect, immune checkpoints and immune checkpoint blockers of gastric cancer, and can be used to evaluate immunosuppressants. Finally, we identified the core gene FAP, which is a biomarker of early gastric cancer and a biomarker of prognosis of gastric cancer. The relationship between FAP and immune checkpoints and immune checkpoint blockers in gastric cancer was established. This discovery can provide a new target for immunotherapy of GC, and provide a new theoretical basis for basic research, clinical diagnosis and individualized treatment of GC. Therefore, our research has great practical value.

  1. Authors should further clarify and elaborate novelty in their contribution.

Thank you for your suggestion. In view of this suggestion, our explanation is that the prognostic risk assessment model composed of necroptosis-related genes(NRGs) in gastric cancer is the best assessment model with relatively few genes so far. Moreover, it is the first time that we used NRGs to screen diagnostic markers for early gastric cancer, and the established 2-NRGs early diagnosis model has only two genes, which also has a good predictive effect. At the same time, we also determined that FAP can be used as a biomarker for early diagnosis and prognosis, Moreover, we also established the relationship between FAP and immune checkpoint inhibitors in gastric cancer. This discovery is the first time to combine NRGs with immunotherapy of gastric cancer, which can provide a new target for immunotherapy of gastric cancer.

  1. Conclusion is too short. Add more explanation.

Thanks for your advice, the conclusion section is indeed too simple.In response to this proposal, we have interpreted in more detail at lines 612-623.

  1. What are the limitations of the present work?

Thank you for your suggestion. In response to this suggestion, our explanation is as follows: First, the data used are retrospective research data from TCGA database and GEO database, which are not forward-looking. Then, the prognostic risk scoring model and early diagnosis model of GC related to necroptosis-related genes were established, which were not verified by clinical tissues. And we used preclinical models to reveal that NRGs may be a biomarker for early diagnosis or prognosis, and analyzed the related research between prognostic risk score and GC patients in immune microenvironment, immune cells and immunotherapy. We need to use relevant samples to verify them at mRNA and protein levels respectively and support our findings. The research on core genes also needs further verification at protein level (See lines 603-610 for details).

This is our reply. We have tried our best to further improve our article and revised it strictly according to the opinions of reviewers. Thank you to Reviewer for your enthusiastic work, hope our correction will be recognized, and thank you again for giving us many valuable suggestions.

Kind regards,

Dr. Zhou

Contributors

E-Mail: [email protected]

Round 2

Reviewer 2 Report

.